# Is Precipitation the Main Trigger of Medium-Magnitude Floods in Large Alpine Catchments?

**Florian Raymond, Bruno Wilhelm * and Sandrine Anquetin** 

Univ. Grenoble Alpes, CNRS, IRD, Grenoble-INP (Institute of Engineering Univ. Grenoble Alpes), IGE, 38031 Grenoble, France; florian.raymond@univ-grenoble-alpes.fr (F.R.); sandrine.anquetin@univ-grenoble-alpes.fr (S.A.)

* Correspondence: bruno.wilhelm@univ-grenoble-alpes.fr

**Abstract:** Flood projections are still highly uncertain, partly resulting from the limited accuracy of simulated precipitation by climate models. To overcome this limitation, recent studies suggest to use direct linkages between atmospheric processes leading precipitation, often better simulated than precipitation, and the flood occurrence. Such an approach implies, however, that historical flood events mainly result from direct contribution of precipitation only. Consequently, this paper has a twofold objective: (i) To explore to what extent the generation of medium-magnitude flood events in a large mountainous catchment can be explained by the precipitation only, and (ii) to identify what are the best features of flood-inducing precipitation episodes (i.e., duration and accumulation). Taking advantage of centennial-long discharge (gauge stations) and precipitation (ERA-20C reanalysis) data series, this study is based on three-year return period flood events of the upper Rhône River (NW European Alps). Our results suggest that half of the studied floods are triggered by precipitation only, but precipitation indices are mainly good only for high-magnitude events with return period of at least 20 years. Hence, modelling flood occurrence directly from atmospheric processes leading precipitation seems to be possible for events with the highest magnitude (i.e., the ones with the highest potential to impact societies).

**Keywords:** high-magnitude floods; historical; precipitation; Alps

## 1. Introduction

Exploring future flood hazard variability is generally accomplished by coupling atmospheric climate projections with land-surface schemes and hydrological models [1]. This practice leads to high uncertainty in the evolution of the flood magnitude and frequency, mainly due to (i) the uncertainties of extreme precipitation projections provided by global and/or regional climate models [2–5], and (ii) the uncertainties resulting from the hydrological modelling [6,7]. To bypass many of these uncertainties, an alternative approach is to rely on direct links between synoptic atmospheric processes and flood occurrences [8,9]. The main driver of flood hazard evolution could be precipitation, which is generated by different atmospheric processes at several embedding scales (e.g., regional pressure distribution, atmospheric humidity, local wind, etc.) [10]. Consequently, the use of atmospheric processes related to extreme precipitation, much better simulated by climate models than precipitation themselves, will indirectly allow the exploration of future changes in extreme precipitation and, thereby, of floods [9]. However, the use of atmospheric indicators to estimate future changes in floods occurrence relies on two main assumptions: (i) historical flood events mainly result from particular "extreme" precipitation accumulation, and that (ii) these extreme precipitation episodes are associated to particular atmospheric features that could be used as predictors from climate projections [9,11].

These assumptions may be questionable in mountainous regions. Indeed, the predictive power of atmospheric indicators related to precipitation may be limited by confounding effect of hydrological processes like, for example, snowmelt and ice or glacier melting. Hence, the main objective of this study is to evaluate to what extent the generation of flood events in a large mountainous catchment is explained by the direct contribution of precipitation only. The second objective is to identify what are the best features of flood-inducing precipitation sequences (i.e., duration and accumulation). If a strong link is found between flood occurrence and precipitation, then the link between precipitation and synoptic atmospheric processes could be explored in a forthcoming study to, ultimately, estimate future trends of flood occurrence. In this study, floods are defined as all events with a daily mean discharge higher than a particular percentile value of the discharge series.

In mountainous hydrosystems, previous studies based on in-situ data series (from weather and gauge stations) starting from the 1960s showed that regular alpine floods (with annual/sub-annual occurrence) are complex events resulting from several types of hydrometeorological processes [12–15]. They particularly show that precipitation is the main process that triggers recurrent events such as flash-floods, short-rain floods, or long-rain floods. They also show that other processes like snow/ice-melt dynamics and soil moisture evolution are associated with recurrent floods, such as rain-on-snow floods, snowmelt floods, or glacier-melt floods. Studies focusing on single flood cases (identified as the largest historical floods) suggest that "extreme" Alpine floods mainly result from anomalous large-scale atmospheric processes that generate heavy precipitation accumulations [16–20]. In line with these findings, we propose to guide our study at the intersection of the two approaches discussed above (focused on numerous regular or single extreme alpine floods), in order to explore whether historical medium-magnitude events are triggered by the direct contribution of precipitation only. The characteristic scales of flood-inducing precipitation (i.e., duration and its accumulation) are explored to highlight the rainy situations for which a direct link between precipitation and medium-magnitude floods is identified. To facilitate this approach, a clustering analysis of the flood events (i.e., flood typology) was conducted, which allowed for studying event (dis)similarities in terms of hydrometeorological processes. Rather than studying the contribution of precipitation for each flood (i.e., individually), the clustering groups together with the events for which (i) the contribution of the precipitation only and (ii) the features of flood-inducing precipitation sequences are comparable. This is performed by studying historical medium-magnitude floods that occurred in a large mountainous catchment (the upper Rhône River, NW European Alps) and using long-time series of discharge and precipitation (almost a century).

Section 2 introduces the studied area and the data used. Section 3 details the three indices used to explore the different flood types. Section 4 compares the main characteristics of the different flood types. Section 5 compares flood types of the Upper Rhône River with those from the literature and discusses the role of the precipitation on each of them.

## 2. Studied Area and Data

### 2.1. The Upper Rhône River Catchment and the Gauge Station of Bognes

The catchment of the upper Rhône River (10,900 km$^2$) is located in the northern French and western Swiss Alps (Figure 1, Table 1). This catchment is under continental climate, with the westerlies bringing moisture from the Atlantic Ocean. At low elevations, mean annual precipitation range from 600 mm (in some parts of Valais, The Switzerland) to 1100 mm (Chamonix, France). In this region, it rains from 30% to 45% of the days, with an annual maximum daily precipitation intensity reaching locally 45 mm/day to 105 mm/day on average [21]. The Bognes gauge station (hereafter referred as Rhône@Bognes, blue star in Figure 1, Table 1) records daily mean discharges at the outlet of the catchment. It is located at Injoux-Génissiat in France, 46 km downstream to the confluence of the Rhône and the Arve Rivers, and 6 km downstream to the confluence of the Rhône and the Valserine Rivers (Figure 1). The hydrological regime of the upper Rhône River at Rhône@Bognes is considered

as glacio-nival, with the lowest and highest monthly mean discharges occurring, respectively, in December (about 270 m$^3$ s$^{-1}$) and July (about 530 m$^3$ s$^{-1}$), for an annual mean discharge of about 358 m$^3$ s$^{-1}$ (Figure A1 in the Appendix A).

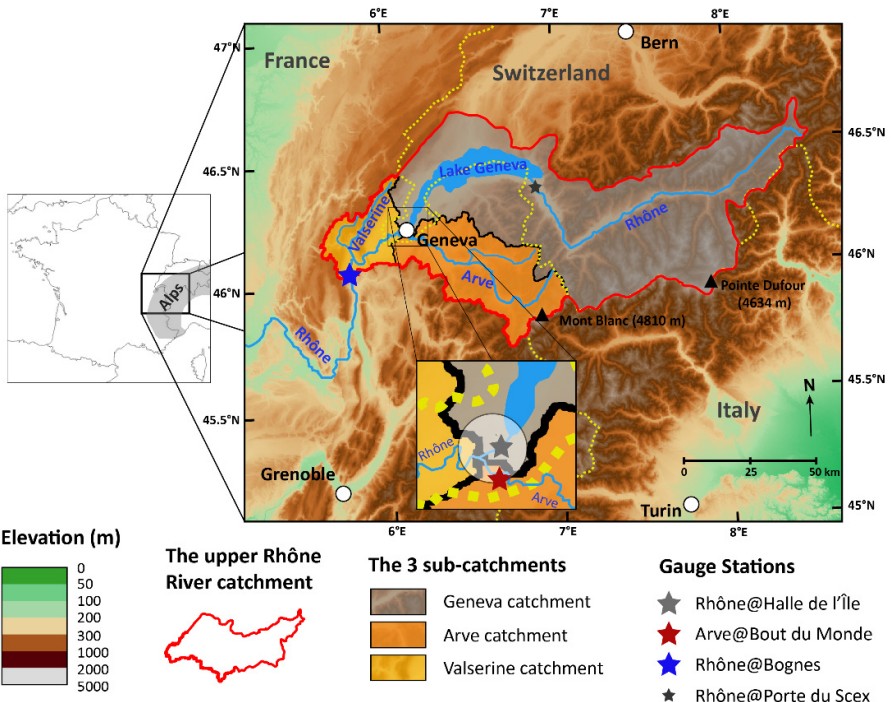

**Figure 1.** Location map of the studied upper Rhône River catchment located in the French and Swiss Alps. The map shows the division of the upper Rhône River catchment into the three sub-catchments as well as the gauge stations used in the study.

**Table 1.** Synthesis of the studied catchments and their associated gauge station. The colors refer to the sub-catchment ones in Figure 1.

| | Name | Surface Area | Gauge Station at the Outlet of the Catchment | | | | |
| --- | --- | --- | --- | --- | --- | --- | --- |
| | | | Station Name | Number | Organization Name | River | Starting Year |
| Main catchment | Upper Rhône River catchment | 10,900 km$^2$ | Rhône@Bognes | 2606 | Federal Office for the Environment (FOEN) | Rhône | 1920 |
| The 3 sub-catchments | Geneva catchment | 8000 km$^2$ | Rhône@HDI | 2170 | Federal Office for the Environment (FOEN) | Rhône | 1904 |
| | Arve catchment | 1900 km$^2$ | Arve@BDM | V1020010 | Federal Office for the Environment (FOEN) | Arve | 1923 |
| | Valserine catchment | 1000 km$^2$ | None, discharge estimated as: Rhône@Bognes - (Rhône@HDI + Arve@BDM) | | | | |

To study the flood dynamics within the upper Rhône River catchment, three sub-catchments, hereafter called Geneva, Arve, and Valserine, were studied and introduced in the following sections.

## 2.2. The Three Sub-Catchments of the Upper Rhône River

The Geneva sub-catchment (8000 km$^2$) corresponds to the Rhône River catchment feeding Lake Geneva (Figure 1, Table 1). It is mainly located in a Swiss high-elevation mountainous area (i.e., mainly the Valais canton but also part of the Vaud canton, mean and maximum altitude of 1660 and 4634 m a.s.l.) characterized by the presence of numerous and large glaciers (covering about 12% of the catchment area). For different reasons (e.g., flood protection, agricultural needs), most of the Rhône River in the Valais was dammed during the nineteenth and the twentieth centuries [22]. In the 1950s, seven dams were built on several Rhône tributaries, mainly for hydroelectric production purposes [23]. The Geneva catchment includes Lake Geneva, the largest lake of Western Europe (covering 580 km$^2$),

mainly fed by the Rhône River coming from the Valais (75% of the lake's water supply) [24]. At the lake outlet, the discharge has been controlled since 1884 to mitigate lake level rises that cause flooding and impact lakefront residents. The gauge station at Halle de l'Île (hereafter referred as Rhône@HDI, grey star in Figure 1, Table 1) is located at the outlet of the Lake Geneva in the city of Geneva and it allows the evaluation of the contribution of the Geneva catchment to the flood generation at Rhône@Bognes. The annual mean discharge at Rhône@HDI is about 250 m$^3$ s$^{-1}$, contributing on average to 70% of the Rhône@Bognes mean discharge. In the Geneva catchment, the discharge of the Rhône River is strongly influenced by ice and snow melting, resulting in a well-marked glacio-nival regime of Rhône@HDI with the highest mean monthly discharges observed in July (about 400 m$^3$ s$^{-1}$ on average; Figure A1 in the Appendix).

The Arve sub-catchment (1900 km$^2$) corresponds to a high-elevation French mountainous area, with a mean and maximum altitude of 1370 and 4810 m a.s.l. (Mont Blanc, highest Alpine summit), respectively. The Mont Blanc massif corresponds to the headwater catchment of the Arve River. The gauge station at Bout du Monde (hereafter referred as Arve@BDM, red star in Figure 1, Table 1) is located in the city of Geneva just before the confluence of the Arve River with the Rhône River and it allows the contribution of the Arve catchment to the flood generation at Rhône@Bognes to be evaluated. The annual mean discharge at Arve@BDM is about 79 m$^3$ s$^{-1}$ and contributes on average to 22% of the Rhône@Bognes mean discharge. The discharge at Arve@BDM is dominated by snow-melt contribution (nival regime) with the highest mean discharges observed in June (about 131 m$^3$ s$^{-1}$; Figure A1 in the Appendix A).

At last, the Valserine sub-catchment (about 1000 km$^2$) includes the Valserine River and several smaller tributaries of the Rhône upstream the station of Rhône@Bognes and downstream Rhône@HDI and Arve@BDM (flowing from the French Jura massif; Figure 1, Table 1). For this sub-catchment, there was no available gauge station. Thus, the contribution of the Valserine catchment to the flood generation at Rhône@Bognes was estimated with Equation (1):

$$QValserine \; = \; QRhône@Bognes - (QRhône@HDI \; + \; QArve@BDM) \qquad (1)$$

where Q refers to the daily mean discharge at the given location.

The mean annual discharge at Valserine is in the range of 30 m$^3$ s$^{-1}$, and contributes on average to 8% of the Rhône@Bognes discharge. The hydrological regime is pluvio-nival (the highest monthly mean discharges occurring in March with about 43 m$^3$ s$^{-1}$; Figure A1 in the Appendix A).

To reduce the influence of the seasonal signature on the hydrological regimes (glacio-nival, nival, or pluvio-nival regimes) in the analysis of the discharge, we used the seasonally adjusted anomalies of the daily mean discharges. These anomalies were computed for each day by comparing the discharge at that day to the mean discharge of all of the corresponding days of the 1923–2010 period. For example, to obtain the seasonally adjusted anomalies for 1 January, we subtracted the average of the 88 1 January data to each 1 January of the period 1923–2010. In the following, we use the term of "discharge anomaly" instead of "seasonally adjusted anomaly" to simplify the text.

The discharge time series at Rhône@Bognes, Rhône@HDI, and Arve@BDM gauge stations were used on the period 1923–2010 that corresponds to the largest common period between both gauge and precipitation time series (Table 1).

### 2.3. The Medium-Magnitude Flood Events

The medium-magnitude flood events of the upper Rhône River catchment were selected based on the 99.9th percentile value (1089 m$^3$ s$^{-1}$) of the daily mean discharge at Rhône@Bognes (1923–2010). This corresponds to select events with return time periods of at least three years. The use of daily discharge series is consistent with the response time of the upper Rhône River catchment (at least one day). Twenty-eight flood events were identified, among them six events were characterized by consecutive days with discharges exceeding 1089 m$^3$ s$^{-1}$. For these latter flood events, the date of the

day with the peak discharge is considered. The first identified event occurred on 19 August 1927 and the last one on 14 January 2004. Among these 28 identified extreme flood events, the flood of February 1990 is the largest event, with daily mean discharge of 1550 m$^3$ s$^{-1}$. The 1990 flood caused a lot of damage in the upper Rhône River catchment, such as the destruction of two bridges in the department of Haute-Savoie (France), traffic interruption on many roads, and the destruction of many houses [25].

*2.4. Precipitation Data*

The time length of the data series from weather stations including daily precipitation accumulation are limited in the French part of the studied area. Indeed, weather stations were mostly installed after the Second World War in the 1950s [26]. The oldest weather stations still active in the French part date back to 1938 and 1944. Since these data do not cover the whole study period (1923–2010), meteorological reanalysis was also used.

2.4.1. Weather Station Data

Insitu daily precipitation observation from 40 weather stations located in and around the Arve, the Valserine, and the Geneva catchments were used for the period 1950–2010 (Appendix C). Since these data represent the actual precipitation that has fallen on the area studied, with a satisfying spatial coverage of data series, they are considered as reference observation series compared to the reanalysis (Section 2.4.2).

The daily catchment precipitation accumulations (referred as the "catchment precipitation" hereafter) were computed based on this data for the same period (1950–2010) by using the Thiessen polygons method [27]. The Geneva catchment precipitation relies on 30 weather stations monitored by MeteoSwiss and Météo-France (green and black dots in Figure A3).

Since the number of weather stations is very low in the Valserine catchment, the mean daily precipitation accumulations from the Arve and the Valserine catchments were considered together (hereafter referred as "A+V catchment precipitation" for a total area of about 2900 km$^2$). The A+V catchment precipitation relies on 16 weather stations monitored by Météo-France and MeteoSwiss (blue and green dots in Figure A3).

The temporal distribution of the A+V catchment precipitation and the Geneva catchment precipitation shows a strong dependence, associated with a significant correlation coefficient of 0.91. It means that, in most cases, when strong precipitation is observed in the Geneva catchment, it is also the case for the A+V catchment.

2.4.2. ERA-20C Reanalysis

The gridded daily precipitation series provided by the ERA-20C reanalysis (1900–2010, 1.125° resolution) [28] were used to cover the entire 1923–2010 period, since it is one of the only daily precipitation datasets covering the entire twentieth century. The daily catchment precipitation accumulations dataset was computed following the same method with the ERA-20C daily precipitation at grid-points present in and around these sub-catchments (Figure A4).

The ERA-20C reanalysis has the advantage of proposing a consequent temporal depth in the data series, including precipitation and variables related on synoptic atmospheric processes possibly leading to the "extremes" precipitation episodes. Nevertheless, these modelled data series present some uncertainties, which require its evaluation from the reference observation series (Section 3, Sections 4.1 and 4.2).

*2.5. Buffer Role of Lake Geneva on Precipitation*

Due to its size and its anthropic regulation, Lake Geneva buffers flood discharges coming from the Geneva catchment [24]. This buffering role and its signature on the discharge time series is explored by using the gauge station of Rhône@HDI (located at the outlet of the lake) and an additional gauge station called Porte du Scex (little dark grey star in Figure 1; called "Rhône@PDS" in Figure A2 Appendix B)

located just upstream of the lake (about 75% of the lake's water supply) [24]. The response time of the catchment was roughly estimated by calculating the correlation coefficient between observation catchment precipitation and discharge time series. Correlation coefficients have been also calculated by shifting the series by one day, two days, etc. The response time of both the "A+V catchment" and at Rhône@PDS (upstream the lake) is estimated to one day since the coefficient correlations are the best when shifting their time series of one day (Figure A2). The response time at Rhône@HDI (downstream the lake) is found to be between three and four days, pointing to the buffering role of Lake Geneva that might influence the flood dynamics at Rhône@Bognes.

To analyze the respective contributions of the different sub-catchments to the flood dynamics at Rhône@Bognes, we compared the percentile values of the discharges at the outlet of the Geneva catchment (Rhône@HDI) and at the outlet of the "A+V catchment" when the discharges at Rhône@Bognes exceed the 99.9th percentile. The results (not shown) point out that the discharge at Rhône@HDI exceeds the 99.9th percentile for only 12% of cases of Rhône@Bognes (18% when using the 99.5th percentile), while the discharge from the "A+V catchment" exceeds the 99.9th percentile for more than 62% of the cases (more than 93% using the 99.5th percentile). This suggests that the discharges flowing from the "A+V catchments" play a dominant role for the generation of floods at Rhône@Bognes, while the contribution of Rhône@HDI is buffered by the lake. Consequently to this strong buffering role of Lake Geneva, we focused on precipitation falling in the A+V catchments to characterize the hydrometeorological processes that triggered the three-year return period flood events at Rhône@Bognes.

## 3. Methods

Identifying features of flood-inducing precipitation events is first required to explore to what extent the generation of three-year return period flood events can be explained by precipitation only. The use of spatial and temporal characteristics of daily pre-floods precipitation (e.g., time of peak concentration, antecedent rainfall, maximum precipitation intensity) as classification criteria can add information about the events' (dis)similarity, resulting in a more robust detection of flood types possibly induced by precipitation only [29].

### 3.1. Descriptors of Precipitation

Different precipitation sequences (based on the catchment precipitation) occurring prior to the flood events were tested to identify the most relevant features of the precipitation events associated to the generation of Rhône River flooding. Two variables were considered to characterize the precipitation sequences: (i) the temporality of the sequence with respect to the flood day (which correspond to the time lag between the end of the precipitation sequences and peak discharge; i.e., sequence ending the flood day, one day before, two days before, etc.), and (ii) the sequence duration (precipitation accumulation during a number of consecutive days). The tested sequence temporality ranged from 10 to 0 days prior to the flood day (i.e., 11 different temporalities of the precipitation sequence with respect to the flood day). The tested sequence duration ranged from 1 to 10 consecutive days (i.e., 10 different durations of precipitation sequence). For example, a precipitation sequence with a duration of 7 days and a temporality of 3 days corresponds to a 7-day sequence ending at D-3 (i.e., a precipitation accumulation lasting 7 consecutive days between the ninth (D-9) and the third (D-3) days prior to the flood day (D)).

Then, the precipitation accumulation of all sequences and their respective mean percentiles were calculated. Accumulation numbers from the reanalysis may be highly uncertain. This is why percentile values were used instead of accumulation values, allowing the events associated with the strongest values in their own distribution of the studied dataset to be considered.

Figure 2 displays the mean percentile values of catchment precipitation accumulation for different sequence durations (color lines) and for different sequence temporality (Up to . . . ). The mean percentiles values were firstly calculated for the 16 floods covered by the observation dataset (1950–2010; Figure 2a)

and secondly for the 28 events covered by the reanalysis dataset (1923–2010; Figure 2b). The 16 floods covered by the observation correspond to the floods 13 to 28 covered by the ERA-20C, the first 12 events covered by the ERA-20C having occurred before 1950.

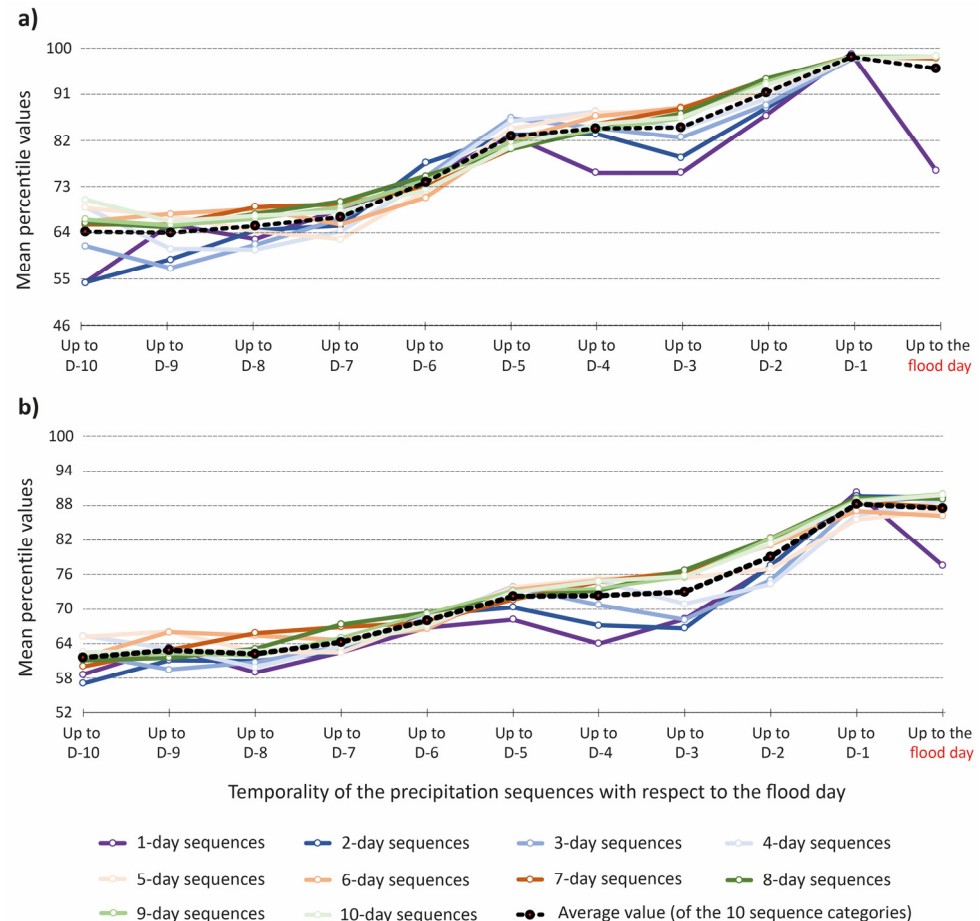

**Figure 2.** Mean percentile values for catchment precipitation accumulation from different sequences being characterized by ten durations (from 1 to 10 consecutive days) and eleven temporalities (ending between 10 and 0 day prior to the flood day) from the A + V catchment precipitation. (**a**) Calculated for the 16 flood events (period 1950–2010) from the observation data series; (**b**) calculated for the 28 flood events (period 1923–2010) from the ERA-20C reanalysis data series. Each colored line corresponds to a sequence with a given duration. The average percentile value of all sequences together is shown in black.

In terms of the temporality of the sequences with respect to the flood day, Figure 2a,b shows very similar trends in percentile values despite slightly larger values for the observation data. It clearly highlights that the percentiles values are the highest for precipitation sequences ending one day prior to the flood, whatever the sequence duration. This sequence temporality is in agreement with the response time of the "A+V catchment" of around one day and also corresponds to Froidevaux et al. (2015) [30] findings for similarly large Swiss catchments (1500–12,000 km$^2$). The precipitation sequences that goes up to one day before the three-year return period flood events seem; therefore, to be the most relevant to explain the link between high precipitation accumulations and flooding.

After having detected the most relevant temporality of the precipitation sequences (ending one day prior to the flood), we focused on the sequence durations. To highlight the sequence durations that have the greatest influence on the flood occurrence, Figure 3 shows the distributions of the percentile values for each of the 10 precipitation sequence durations (i.e., duration from 1 to 10 consecutive days, all ending the day prior to the flood date) calculated from observation (Figure 3a) and reanalysis

(Figure 3b) datasets. Sequences with the lowest medians and dispersions of precipitation accumulation percentiles were clearly characterized by both short (1 and 2 days) and long (6 to 10 days) durations (Figure 3). This result is detected based on both datasets (observation and ERA-20C reanalysis) with slight differences (e.g., the number of days between 7 (Figure 3a) and 8 (Figure 3b) for the long duration). This suggests that both precipitation of short and long duration may result in Rhône River flood occurrence. Two durations were kept to respectively characterize short and long sequence durations. These two durations will then be used as independent indicators.

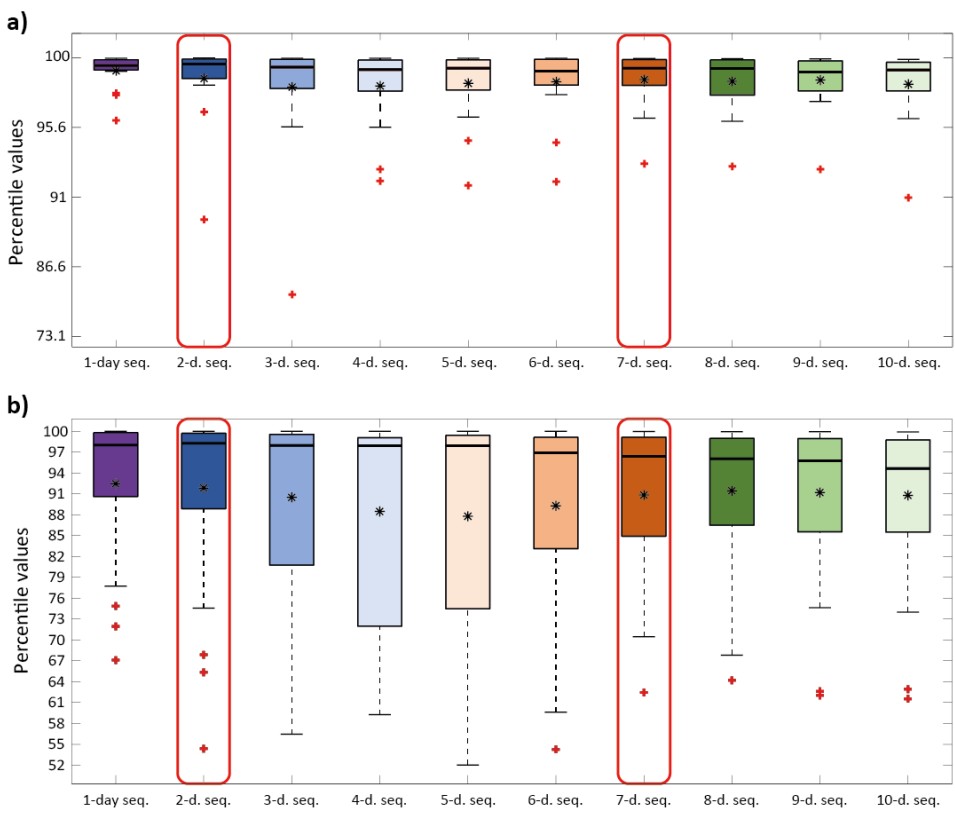

**Figure 3.** Plots (first and third quartiles) of the percentile values for each of the 10 precipitation sequence durations, all ending the day prior to the flood. The black bold bar inside the box corresponds to the median and the black star to the mean. Red crosses indicate extreme values, the light horizontal bars represent the lowest data within the 1.5 inter-quartile range of the lower quartile, and the highest data still within the 1.5 inter-quartile range of the upper quartile. (**a**) For the 16 flood events covered by the observations; (**b**) for the 28 flood events covered by the ERA-20C reanalysis. The red rectangles correspond to the precipitation descriptors selected to perform the flood typology.

For the short sequences, distributions of sequences lasting 1 and 2 days were very similar. Duration of 2 days was then preferred, since it may make it easier to find direct links between synoptic atmospheric processes and flood occurrences. Indeed, links between multi-day precipitation and synoptic meteorological systems are expected to be stronger. In addition, this characteristic duration was in line with the results of Froidevaux et al. (2015) [30], highlighting that precipitation accumulation between 3 to 0 days before the flood is the most relevant factor for flood generation in large catchments of the Alpine area. Regarding the distributions of the long sequence (6–10 days), the 7-day precipitation sequences show the highest mean or/and median percentile values and display the lowest dispersion in Figure 3a, but also the highest median values for the long sequences in Figure 3b. Thus, the duration of 7 days was considered to characterize the long sequences.

Hence, the two precipitation descriptors used to perform the flood typology were: (i) Two-day precipitation accumulation prior to the flood event (i.e., accumulation from D-3 to D-1) and (ii) 7-day precipitation accumulation prior to the flood event (i.e., accumulation from D-7 to D-1).

### 3.2. Descriptor of Discharge Variability

High water levels were sometimes observed many days prior to the flood events and this often happens over longer periods than the 7 days covered by the precipitation descriptors. Therefore, a complementary descriptor to the precipitation ones based on the daily mean discharge variability at Rhône@Bognes before the flood events is introduced. This descriptor is a variation coefficient (VC, Equation (2)) that allows taking into account such long-term high water stages preceding the rise of the water level to the flood peak:

$$VC_{(from\ D-6\ to\ the\ flood\ day)} = \frac{\sigma}{\bar{x}}, \tag{2}$$

where VC is the daily mean discharge variation coefficient at Rhône@Bognes from D-6 to the flood day, $\sigma$ the standard deviation of the daily mean discharge between D-6 and the flood day and $\bar{x}$ the average discharge between D-6 and the flood day. VC was computed from D-6 to the flood day to consider the 1-day response time of the "A+V catchment" to the 7-day precipitation sequences. VC was used as the third descriptor to perform the flood typology.

## 4. Clustering and Resulting Flood Typology

The clustering analyses of the studied flood events allows information about the events' (dis)similarity in terms of precipitation contribution to be added. This section details the flood type clustering based on the three descriptors defined previously. The clustering was also performed with the two data sets: the observation and the ERAC-20C data series.

### 4.1. Hierarchical Clustering

The hierarchical ascendant classification algorithm [31] uses the three above-mentioned descriptors to identify if different flood types exist and if so how many. This algorithm tends to group individuals according to a similarity criterion expressed in the form of a matrix of distances (Euclidean distance metric here), using three-dimensional space defined by the three selected descriptors. It expresses the distance existing between each individual taken two by two [32]. The objective of this method is to divide a population into different classes by minimizing intra-class distance and maximizing inter-class distance. Given the different number of events to be classified (i.e., 16 for the 1950–2010 period; 28 for the 1923–2010 period), respectively, three and four classes are retained from the hierarchical clustering algorithm because these partitions is the best compromise displaying the greatest relative loss of intra-class distance and gain of inter-class distance.

### 4.2. Hydro-Meteorological Characteristics of the Flood Types

Figure 4 shows the flood types obtained with the two datasets. The flood type 1 (1950–2010 period; Figure 4a) groups six events characterized by (i) a zero discharge anomaly until D-2, (ii) high precipitation accumulations from D-2 to D-1, and (iii) a fast and large increase of discharge from D-1 to the flood peaks (+930 m$^3$ s$^{-1}$ on average, the highest of all the types). Aflood type with comparable characteristics is detected for the 1923–2010 period with five events (Figure 4b). The flood type 1 identified from both datasets have three events in common. In both cases, the inter-event discharge variability is very low when comparing with the others flood types.

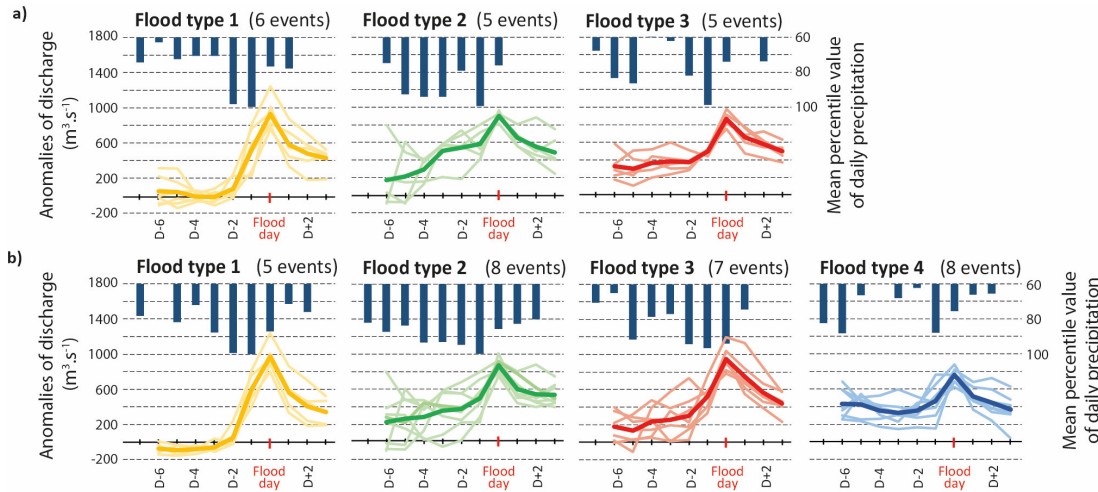

**Figure 4.** Mean hyetograms and seasonally-adjusted hydrographs (the thin line represents each flood event and the bold line represents the mean) associated to each flood type: (**a**) Based on observations that cover 16 flood events (1950–2010); (**b**) based on the reanalysis that covers 28 flood events (1923–2010).

The flood type 2 (1950–2010 period; Figure 4a) groups five events characterized by (i) a regular and large increase of discharge, (ii) a similar increase in daily precipitation accumulations, and (iii) a high precipitation accumulations at D-1 leading to a mean flood peak discharge anomaly near to +900 $m^3 \ s^{-1}$. A flood type with comparable characteristics is detected for the 1923–2010 period with eight events (Figure 4b). The flood type 2 identified from both datasets have three events in common.

The flood type 3 (1950–2010 period; Figure 4a) groups five events characterized by (i) a positive but regular discharge anomaly around until D-2, (ii) significantly high precipitation accumulation at D-1, and (iii) a mean peak discharge anomaly about +900 $m^3 \ s^{-1}$. A flood type with comparable characteristics is detected for the 1923–2010 period with seven events. One event is common between the two classifications for type 3.

For the 1923–2010 period, a flood type 4 gathering eight events is detected and characterized by (i) very high and stable discharge anomalies from D-6 to D-2, (ii) low-to-moderate precipitation accumulations, and (iii) a mean peak discharge anomaly about +820 $m^3 \ s^{-1}$ (the lowest of the four types; Figure 4b). Among the eight events associated with this type, six of them occurred before 1950 (Figure 5a) and are, thus, not considered in the 1950–2010 clustering.

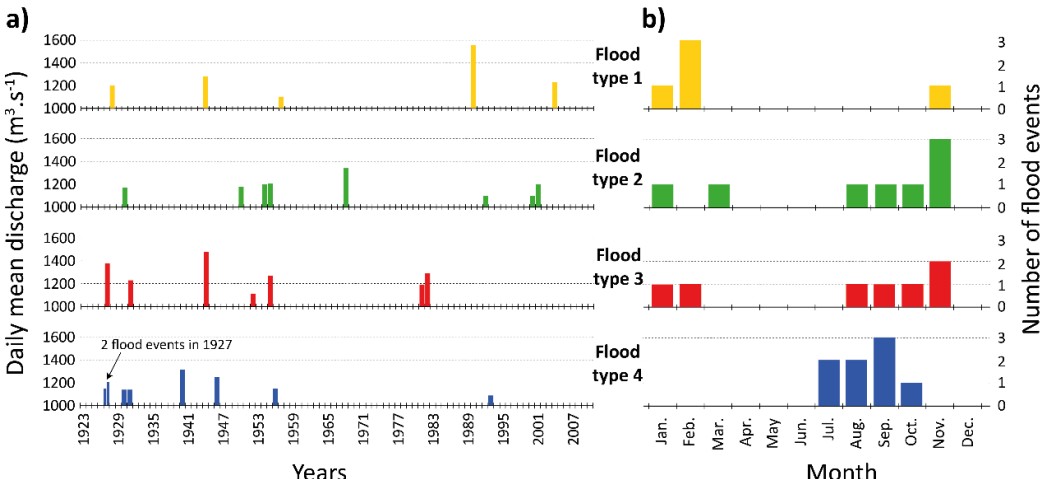

**Figure 5.** (**a**) Chronology (with daily mean discharge values) and (**b**) seasonality of the 28 flood events by flood type.

The weak differences between the flood types identified from the two datasets are mainly due to the different size of the sample considered, respectively, since type 4 seems to highlight a type of flood poorly represented in the period covered by the observation. Except type 4, this comparison overall highlights that the flood typology seems independent from the precipitation data used, supporting the added value of the reanalysis to document periods not covered by observations. This finding supports the use of ERA-20C data to further explore to what extent the generation of the 28 three-year return period flood events identified during the period 1923–2010 can be explained by precipitation only.

*4.3. Temporal Characteristics of the Four Flood Types*

Flood types 1, 2, and 3 occurred mainly in autumn and winter (Figure 5b). They are distributed over the whole period (1923–2010) without any clear cluster that would reflect flood-rich/flood-poor periods (Figure 5a). Regarding the flood magnitude, type 1 and type 3 include the largest flooding; in February 1990 (1550 $m^3$ $s^{-1}$) for type 1 and September 1927 (1380 $m^3$ $s^{-1}$) as well as November 1944 (1480 $m^3$ $s^{-1}$) for type 3. Beyond these largest events, no clear change in flood magnitude can be noted since floods with the highest magnitude are identified at both the beginning and the end of the studied period.

By contrast, flood type 4 occurred in summer and beginning of fall (Figure 5b). In addition, only one of the eight events occurred after the 1960s (moreover the weakest flood of the 28 events). To understand the absence of this flood type from the 1960s, a homogeneity test of Pettitt [33] was applied to the daily discharge series in summer and beginning of fall at gauge stations of Rhône@Bognes, Arve@BDM, Rhône@HDI (downstream of lake Geneva), and Rhône@PDS (upstream of the lake). A break is detected in August 1961 at Rhône@Bognes (at the 0.95 confidence level) with a decrease of 17% of the mean daily discharge afterwards. No break is found in Arve@BDM, while a break is also detected in August 1961 at Rhône@HDI and in July 1957 in Rhône@PDS, with a similar decrease of 16% to 20%, respectively, of the mean daily discharge. Finding a break relatively synchronous between the stations and in a similar range of discharge, the absence of type 4 from the 1960s may result from this decrease in discharge upstream of Lake Geneva. Since the change abruptly occurred, the trigger is more likely related to changes in river management than climate [8]. Thereby, the break at Rhône@Bognes seems to be strongly related to discharge changes in the Valais catchment, where seven dams were built on tributaries of the Rhône River at that time [23]. These dams aim to store glacially-fed waters in summer and to release them mainly in winter for hydroelectricity production, when natural discharges are low and energetic needs are high (e.g., heating).

## 5. Discussion: Precipitation Only vs. Hydrological Processes for Flood Triggering

In this discussion section, we firstly summarize the main characteristics of the different flood types previously discussed from literature. Secondly, a comparison between the four flood types detected in this study and those from the literature is done. Finally, the role of the precipitation is discussed for the different flood types, but also for the different magnitudes of floods, regardless of the type.

*5.1. Alpine Flood Types from the Literature*

Previous studies show that recurrent floods in the Alps are complex events resulting from mixed meteorological and hydrological processes [12–15]. The key characteristics of the different flood types identified by these studies are summarized in Table 2.

**Table 2.** Main characteristics of the Alpine flood types from the literature. "RR" means "precipitation". Based on Table 2 from Tarasova et al. (2019).

| Reference | Studied Area, Period, and Return Period of Events | Descriptors | Flood Types | | | | | Glacier-Melt |
|---|---|---|---|---|---|---|---|---|
| | | | Rainfall-Induced | | | Snowmelt-Induced | | |
| | | | Flash Flood | Short-Rain | Long-Rain | Snow-Melt | Rain-On-Snow | |
| Merz and Blöschl (2003) [12] | Total of 490 catchments; 3–30,000 km$^2$; Austria; 1971–1997; sub-annual events | Date; one- and three-day rainfall volume; snow water equivalent and snowmelt; runoff coefficient; time of concentration | Summer; RR <90 min duration; very intense response; area <30 km$^2$ | No seasonality; RR of one day duration; fast response; local or regional extent | No seasonality; RR > one day duration; slow response; spatial extent >10$^4$ km$^2$ | Spring and summer; no rain needed; medium or slow response; medium spatial extent of floods | Between cold and warm periods; at least moderate rainfall events; from fast to slow response; limited to catchments with snow cover | |
| Sikorska et al. (2015) [13] | Nine catchments; 2–939 km$^2$; The Switzerland; 1981–2012; sub-annual events | Date; rainfall intensity; volume and duration; snow water equivalent and snowmelt; glacier melt; antecedent soil moisture | Summer–autumn; RR < half day; local area | No seasonality; RR of maximum one day duration; local or regional extent | No seasonality; RR > one day duration; regional extent | Possibly during the whole year; no rain needed | Possibly during the whole year; rainfall events with at least moderate intensity; limited to catchments with snow cover | Mostly during summer; no rain needed; limited to glaciated catchments |
| Brunner et al. (2017) [14] | Total of 39 catchments; 20–1700 km$^2$; Switzerland; 1961–2013; sub-annual events | The same as Sikorska et al. (2015) [13] | RR < half day | RR of maximum one day duration | RR > one day duration | No rain needed | At least moderate rainfall events; limited to catchments with snow cover | Mostly during summer; limited to glaciated catchments |
| Keller et al. (2018) [15] | One catchment of 1702 km$^2$; Switzerland; 1961–2014 annual floods | Rainfall duration; rainfall volume;95th quantile of spatial rainfall distribution; snow cover | High intensity and sums; high intensity | | High sums | Rain and snow (if snow cover area >40% of catchment area);low intensity and low sums | | |

Three flood types are mainly triggered by precipitation: (i) the flash floods, (ii) the short-rain floods, and (iii) the long-rain floods (Table 2). The flash floods type characterizes events occurring mainly in summer in small catchments due to short (less than a half day) but very intense precipitation of convective origin [12,13]. The short-rain type characterizes events triggered by intense precipitation lasting around one day. These events could have a local to regional extent and have no particular seasonality [12–14]. In Keller et al. (2018) [15], these two types are close to the types called "high intensities–high sums" and "high intensities". The long-rain type characterizes events resulting from large precipitation accumulation falling during several days (including low intensity precipitation). These events have a regional extent and no particular seasonality [12–14]. In Keller et al. (2018) [15], this type is close to the "high sums" type.

In addition, three other flood types are highlighted from these studies (Table 2). The rain-on-snow type characterizes events resulting from precipitation (at least moderate accumulation) falling on an existing snow cover (about max of 1000 km$^2$). The snowmelt type is caused by snowmelt during warm fair weather, possibly during the entire year, but mainly during spring and summer. The glacier-melt type characterizes events caused by high glacial melting due to air warming in glaciated catchments. The role of precipitation as flood trigger for the two last flood types is null to weak.

*5.2. Comparison with the Flood Types of the Literature*

Are the flood types detected in this study (based on at least three-year return period events detected from 1923 and only based on daily precipitation and discharge data series) comparable to the flood types previously detected in the literature (based on recurrent events from the second part of the twentieth century and based on numerous hourly time step meteorological and hydrological series)?

Flood type 1 mainly results from relatively short and heavy precipitation sequences (high precipitation accumulation from D-2 to D-1; (Figure 6a,b)). During the flood day, the contribution of the Geneva catchment is very low, while the ones of the Arve and the Valserine catchments reach respectively values higher than 42% (Figure 6b). For example, during the flood of February 1990, the peak discharge of the Valserine River reached 360 m$^3$ s$^{-1}$ (value estimated higher than a 50-year return period discharge) [34], while the Geneva catchment played a weak role due to the regulation and/or a slower response to this heavy precipitation. This precipitation duration and the regional extent are similar features to the "short-rain floods" type of the literature (Table 2). The duration of heavy precipitation of type 1 (two days) is slightly longer than in the definition of the previous studies (one day). This difference may result from the larger size of the studied catchment compared to those of the literature.

Flood type 2 is associated to a combination of both long and short and heavy rain episodes, triggering a regular increase of discharge until the flood peak (Figure 6a,b). The high discharges in Rhône@Bognes from D-6 to D-1 are mainly provided by the Geneva and Arve catchments (Figure 6b). The contribution of the Arve catchment from D-1 to the flood day explained 45% of the flood events (Figure 6b). Thus, flood type 2 is very similar to the "long-rain floods" type defined by Merz and Blöschl (2003) [12], Sikorska et al. (2015) [13], and Brunner et al. (2017) [14], as those events are triggered by (i) rainfall over several days that saturates the catchment and cause high discharge conditions, and (ii) additional heavy rainfall that generates the flood peak. Compared to Keller et al. (2018) [15], flood type 2 is closed to their "long duration floods" characterized by high precipitation depths and embedded episodes of high precipitation intensities.

Flood type 3 is mainly triggered by two-day high precipitation accumulation falling from D-2 to D-1 when the discharge is already high. However, cumulative precipitation from D-7 to D-3 are lower than those of flood type 2 (Figure 6a) and cannot explain alone the high discharge at D-2 (Figure 6b). This suggests that hydrological processes play a role from D-7 to D-2. Among them, ice melting is unlikely in this season (autumn–winter) and soils are expected to be wet and saturated since this season is rather wet and cold. Consequently, (early) snowmelt is then the most probable candidate since a large part of the catchment may be covered by snow and sensitive to changes in temperature

and rainfall. Therefore, flood type 3 seems to result mainly from mixed hydrological (e.g., snowmelt) and meteorological (i.e., short intense precipitation sequence). The role of snowmelt cannot be further explored due to the lack of data on snow cover covering the whole studied period. The precipitation characteristics makes this flood type 3 similar to the "short-rain floods" (Table 2) or to the flood type defined as "shorter duration events with higher precipitation intensity" by Keller et al. 2018 [15]. These types; however, do not include a strong hydrological component such as snowmelt. The type 3 could, thus, be an intermediate case between the "short-rain floods", the "rain-on-snow floods", and "snowmelt" of Merz and Blöschl (2003) [12].

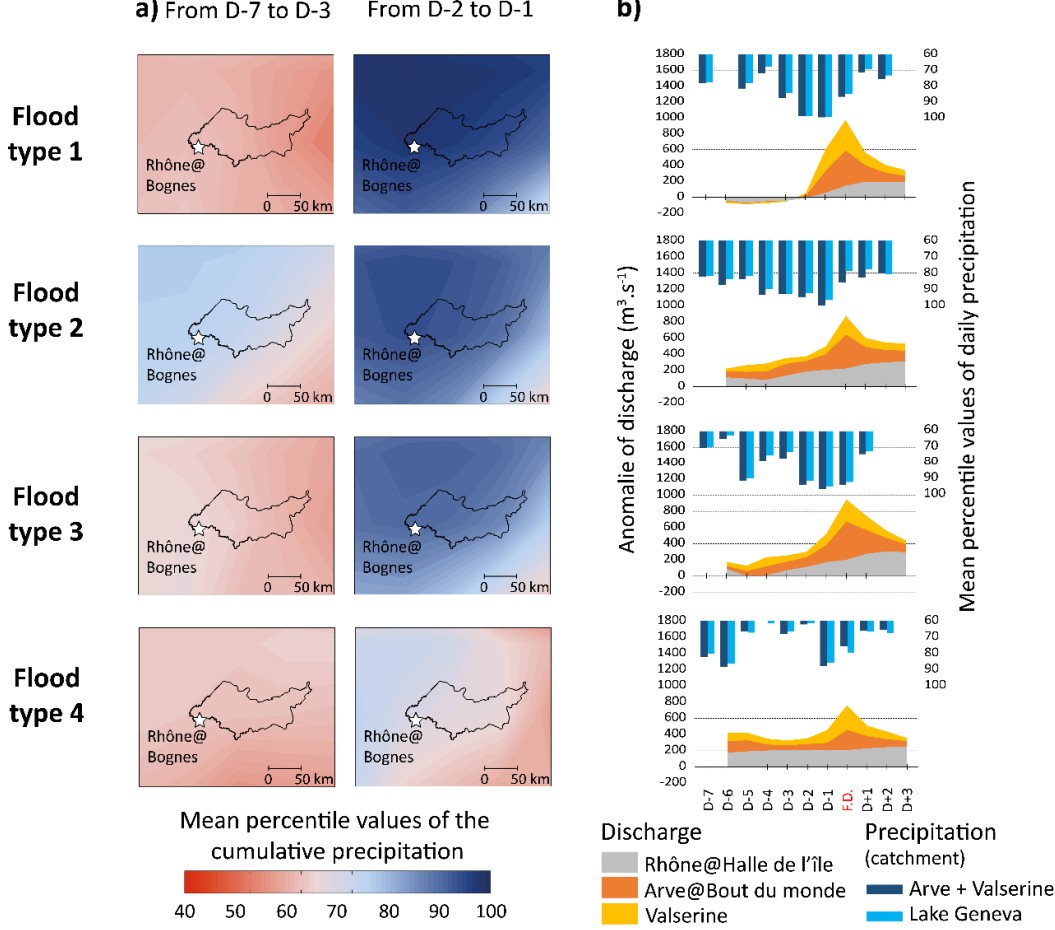

**Figure 6.** Each of the four flood types of the period 1923–2010: (**a**) Map of the mean percentile values of the cumulative precipitation sequences; (**b**) hydrographs and hyetograms associated to the flood types. The mean percentile values of the cumulative precipitation are given for the global periods D-7 to D-3 and D-2 to D-1. The hyetograms show the mean daily percentile values of daily precipitation for the Arve + Valserine (A + V) and Geneva catchment precipitation. The hydrographs show the daily mean seasonal adjusted discharge anomaly of discharge for each sub-catchment. The accumulation of the three anomalies give the daily mean seasonal adjusted discharge anomaly observed in Rhône@Bognes.

Flood type 4 occurs in summer and fall and is associated with moderate precipitation. Thereby, this type cannot be explained by precipitation only (Figure 5). A detailed analysis of the respective contribution of the three catchments reveals that the Geneva catchment plays a dominant role, by triggering high discharge anomalies before the flood events that correspond to more than 50% of the total discharge anomalies from D-6 to D-2 (Figure 6b). To better understand the reason of these high and long-lasting discharge anomalies, the Geneva catchment precipitation was also analyzed. As said in Section 2.4.1, the Geneva catchment precipitation is very similar to the A+V catchment precipitation

(correlation coefficient of 0.91) and, thereby, they cannot explain these high and long-lasting discharge anomalies provided only by the Geneva catchment. In addition, for five of the eight type 4 events, high discharge anomalies coming from the Geneva catchment lasted 20 to 120 consecutive days. Such high water stages in (late) summer may then result from particularly intense melting of the numerous and large glaciers of the Valais and part of the Vaud cantons, since some monitored glaciers (e.g., Mer de Glace and Argentière glaciers) have been affected by important ice losses during the summers that the flood type 4 events occurred [35,36]. These glaciers are located a few tens of kilometers from the Valais and the Vaud cantons, but ice losses are expected to be similar at the regional scale [37]. These findings support a glacial trigger of the high, long-lasting summer discharge anomalies that made flooding possible without heavy precipitation. Soil saturation may also influence the generation of flood type 4. Indeed, a precipitation sequence occurred a few days before the flood event (around D-6) triggering only a slight increase of discharge, while a second episode (D-1) with similar precipitation accumulation led to the flood peak (Figure 6b). This suggests that soil infiltration buffer the first precipitation event, while soils are saturated at the time of the second precipitation event, promoting runoff and, thereby, the occurrence of three-year return period flood events. On another hand, the increase of discharge results from the contribution of the Arve River and in a larger part from the contribution of the Valserine River that flows from the Jura massif, an area where soil saturation has been recognized as a key process for flood generation [30]. Such a role of soil saturation is not as clearly identified for the other flood types. This may be related to the seasonality of the other flood types that occurred in late autumn and winter (i.e., during the rainiest and coldest period that makes soils often saturated and that limits soil evaporation). By contrast, the period of flood type 4 occurrence (i.e., summer and beginning of autumn), is rather dry and, thereby, soils are more sensitive to moisture variations. Therefore, flood type 4 seems to result from a combination of (i) intense ice-melting that triggers a high, long-lasting discharge baseline, (ii) a first precipitation event of moderate accumulation lasting a few days (D-7 to D-5) and saturating soils, as well as (iii) a second precipitation event of moderate-to-high accumulation (D-1) resulting in the flood peak. This type 4 is really similar to the "glacier-melt" type from Sikorska et al. (2015) [13] and Brunner et al. (2017) [14]. However, precipitation events are here required to trigger the flood peak, while the "glacier-melt" type from the literature results from glacier melting only.

Finally, the Geneva catchment was first assumed to plays a negligible role on flood generation at Bognes because of the large size of the lake and its regulation that buffer the discharge variability. Nevertheless, the Geneva catchment may contribute to the flood generation by providing high water level downstream over longer time scale than the typical one of flood generation as identified by Froidevaux et al., (2015) [30]. This contributes significantly to flood type 4 and, possibly, in a very lesser extent to flood types 2 and 3.

*5.3. Role of Precipitation in Both Flood Type and Flood Magnitude*

In line with the literature on flood typologies of Alpine rivers, our results suggest two main groups of flood types occur on the upper Rhône River: (i) types 1 and 2 for which direct accumulation of precipitation is the main trigger of three-year return period flood events, and (ii) types 3 and 4 for which such floods result from a combination of precipitation as well as others processes (e.g., possibly ice-melt, soil moisture, snowmelt). Our results also suggest that precipitation sequences triggering flood types 1 and 2 are described by two distinct durations of two and seven days. To highlight the role of those precipitation sequences on the studied flood events, all of them are plotted in a diagram of short (two-day) versus long (seven-day) precipitation sequence percentiles (Figure 7). When precipitation sequences are strongly involved in the generation of a flood event, this event is expected to appear in the uppermost and/or the rightmost part of the diagram. Flood events of types 1 and 2 appear all in the upper right corner of Figure 7 (left panel), while events of type 3 and even more for events of type 4 are scattered throughout the diagram. This strongly supports that types 1 and 2 are mainly resulting from these short and long precipitation sequences. By contrast, this also highlights the minor role of

precipitation in the generation of flood type 4 events. In detail, this diagram confirms that flood type 1 mostly results from short precipitation sequences, while flood type 2 results from various combinations of both short and long precipitation sequences (also shown in Figure 6a). However, a few annual flood events (annual maximum discharge) seem to result from precipitation sequences with similar, or even higher, percentiles than flood types 1 and 2. This may suggest that these precipitation features would not be fully unequivocal indicators of these three-year return period events, limiting the use of these indicators as "perfect" predictors of medium-magnitude flood occurrences. However, this may also result from the uncertainties of the ERA-20C precipitation dataset. Beyond this limitation, predicting three-year return period flood events with these indicators would not be relevant since only types 1 and 2 are well characterized by these indicators (i.e., types 3 and 4 would not be predicted). Therefore, the flood typology helps to identify and characterize the role of precipitation as a flood trigger, but this does not help predict all the medium-magnitude floods.

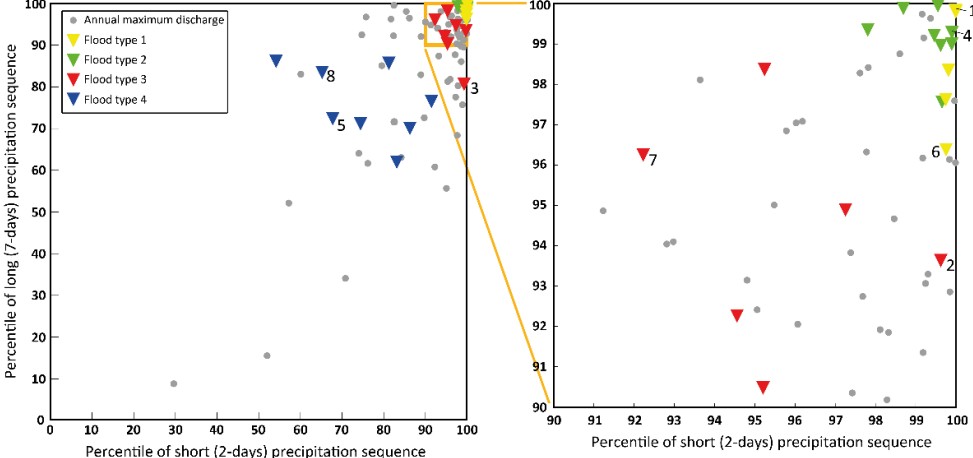

**Figure 7.** Percentiles of short (two-day, x axis) versus long (seven, y axis) precipitation sequences for each flood type, as well as for annual maximum discharge. Numbers show the eight largest floods of the last 88 years (i.e., greater than 10-year return period events). The right-hand panel is a zoom of the diagram for percentile values higher than 90.

Instead, considering the flood types, a view from the flood magnitude can be envisioned. The largest flood event (February 1990, 100-year return period; [38], labelled 1, Figure 7) is characterized by the heaviest precipitation with the most extreme percentile of short precipitation sequence (99.99 with ERA-20C, Figure 7; 100 with observations, not shown). The next three largest events (i.e., 20-year return period events (labelled 2 to 4, Figure 7)) also result from very high percentiles of short precipitation sequences (>99.2), while percentiles of long precipitation sequences range from 80.9 to 99.3. Thereby, rain accumulations of short precipitation sequences seem to be much more relevant to trigger those floods than those of long precipitation sequences. Regarding the next four events in order of flood magnitude (i.e., the 10-year return period events (labelled 5–8, Figure 7)), only the flood event with the sixth highest magnitude is associated with a very high percentile (>99) of short precipitation sequence. Therefore, 10-year return period events cannot be systematically attributed to such short, heavy precipitation sequences. The two precipitation indices appear to be less and less relevant when considering all flood events, even more when considering annual maximum discharges (grey dots in Figure 7). This suggests that considering floods of weaker magnitude progressively shows a decreasing role of precipitation accumulations, suggesting a higher diversity of involved processes in the generation of, for example, annual flooding. This result is relatively similar to results of Merz and Blöschl (2003) [12], suggesting a dominant role of precipitation to trigger >10-year return period flood events. Overall, this suggests that high-magnitude flood events (i.e., with a return period of at

least 20 years) result all from large precipitation accumulation over two days. Hence, the use of this precipitation indicator seems to be relevant for predicting the occurrence of the largest flood events.

## 6. Conclusions

The main objective of this study was to explore to what extent the generation of medium-magnitude flood events in a large mountainous catchment can be explained by precipitation only and, if so, what are the features of the precipitation. This objective is a prerequisite to develop a predictive model of medium-magnitude flood occurrence based on atmospheric processes directly (that would be associated to the precipitation sequences discussed in this study) instead of the classical model-chain approach. A flood typology of at least three-year return period events occurring between 1923 and 2010 on the upper Rhône River was performed through three indices based on precipitation (two-day and seven precipitation accumulations) and discharge (variation coefficient) series. This resulted in four flood types:

- Winter type 1 results from a heavy short precipitation sequence (Figure 7), corresponding well to the "short-rain floods" type with a duration of heavy precipitation (two days) longer than in the definition (e.g., Merz and Blöschl, 2003 [12]).
- Autumn/winter type 2 results from the combination of short and long intense precipitation sequences, similar to the "long-rain floods" type defined by Merz and Blöschl (2003) [12].
- Autumn/winter type 3 seems to result mainly from both short and intense precipitation as well as others processes such as snowmelt.
- Summer type 4 resulting from a combination of (i) intense ice-melting that triggers a high, long-lasting discharge baseline, (ii) a first precipitation event of moderate accumulation lasting a few days (D-7 to D-5) and saturating soils, as well as (iii) a second precipitation event of moderate-to-high accumulation (D-1) resulting in the flood peak.

Thus, the typology highlighted that only half of the three-year return period floods (i.e., 13 of the 28 flood events) seem to result from the direct contribution of intense precipitation accumulation only. In addition, a few annual flood events seem to result from similar precipitation, suggesting that the identified precipitation sequences would not be "perfect" predictors of medium-magnitude flood occurrence. However, a detailed analysis focusing on flood events with the largest magnitude showed that the 20-year return period events (high-magnitude events) (i.e., those with the largest potential to impact societies) all result from precipitation characterized by large accumulations over two days. Hence, our result suggests that predictive models of high-magnitude flood occurrence based on atmospheric processes directly could be successfully developed on such a large, mountainous catchment by looking for unequivocal atmospheric predictors of heavy, two-day-long precipitation.

To find relevant atmospheric predictors of the short and heavy two-day-long precipitation sequences, it could be interesting to focus, for instance, on the integrated water vapor transport flux (called IWT, integrating the specific humidity and the zonal and meridional wind components) [39]. This atmospheric component would allow important wet flows coming from the west, which could result to such short and heavy precipitation sequences in contact with the Alps, to be detected. Another interesting atmospheric process to be analyzed would be the potential vorticity (PV) [40]. The PV analysis would permit the detection of PV-streamer (also called Rossby waves-breaking; [41]), which may be robust precursors of heavy precipitation events in the Alps, such as the ones that led to the abundant precipitation triggering the extreme flood event in Lago Maggiore catchment in 1868 [20]. This opens a promising avenue for complementary flood hazard projections if robust atmospheric predictors are found.

**Author Contributions:** All of the authors helped to conceive and design the analysis. F.R. performed the analysis and B.W. and S.A. participated in the analysis. F.R., B.W. and S.A. wrote the manuscript.

**Funding:** This research received no external funding.

**Acknowledgments:** This work is a contribution to the Cross Disciplinary Program "Trajectories", from the Grenoble University. Within the CDP-Trajectories framework, this work is supported by the French National Research Agency in the framework of the "Investissements d'avenir" program (ANR-15-IDEX-02). The authors are grateful to the Compagnie Nationale du Rhône (CNR) and to the Federal Office for the Environment of Switzerland (FOEN) for providing discharge measurement data series from gauge stations located in France and Switzerland. The authors are also grateful to the MeteoSwiss and to the Météo-France organizations for providing observation precipitation data series from meteorological stations located in France and Switzerland.

**Conflicts of Interest:** The authors declare no conflicts of interest.

## Appendix A

Monthly mean hydrographs of the Rhône@Bognes and for the three sub-catchments, during the 1923–2010 period.

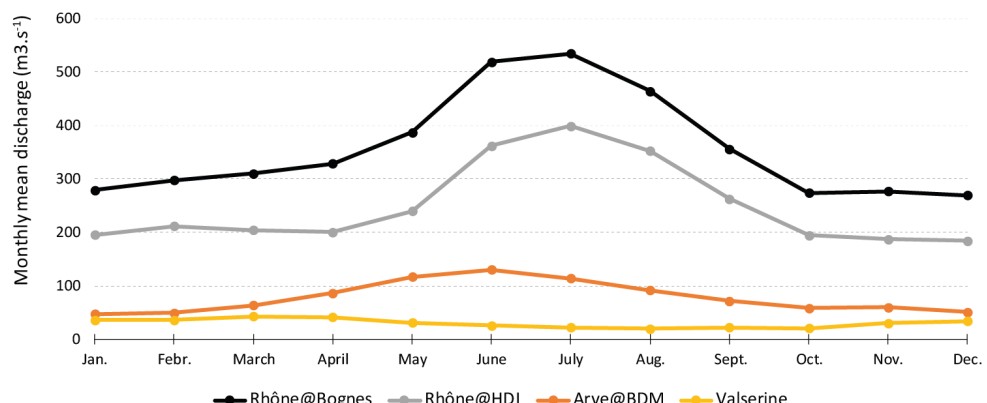

**Figure A1.** Monthly mean hydrographs of the discharge series for the Upper Rhône River catchment (Rhône@Bognes) and the three studied sub-catchments (Rhône@HDI, Arve@BDM, and Valserine), during the 1923–2010 period.

## Appendix B

Response time of the A+V and Geneva catchments.

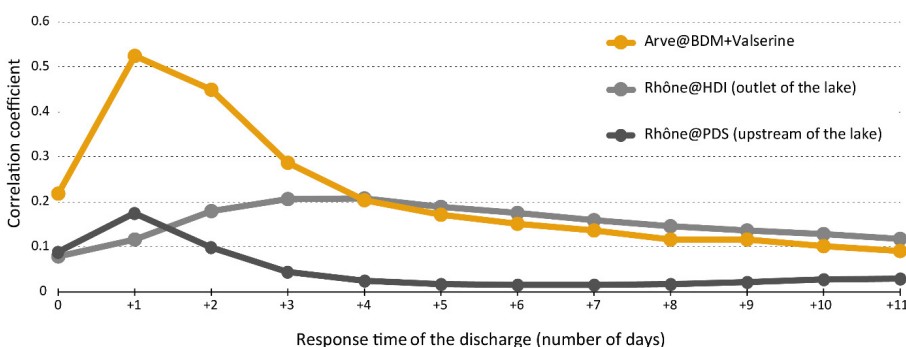

**Figure A2.** Correlation coefficient between the precipitation and discharge time series for the 1950–2010 period. The catchment precipitation is based on the observations. For the calculation of the correlation coefficient à D+1, D+2, etc., the discharge time series are shifted by one day, two days, etc.

## Appendix C

Data series used for the calculation of the daily catchment precipitation.

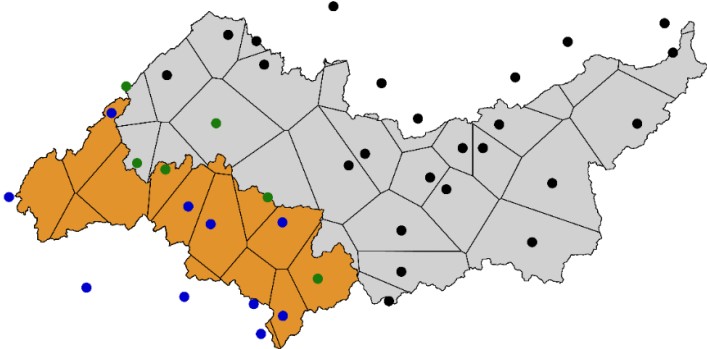

**Figure A3.** Location of the different weather stations used to calculate the observed catchment precipitation for the period 1950–2010. The orange area corresponds to the A+V catchment, the green and blue dots correspond to the 16 weather stations used to calculate the catchment precipitation (the Thiessen polygons associated are shown). The grey area corresponds to the Geneva catchment, the black and the green dots correspond to the 30 weather stations used to calculate the catchment precipitation (the Thiessen polygons are shown). Green dots represent weather stations both used for the calculation of the A+V and Geneva catchment precipitation.

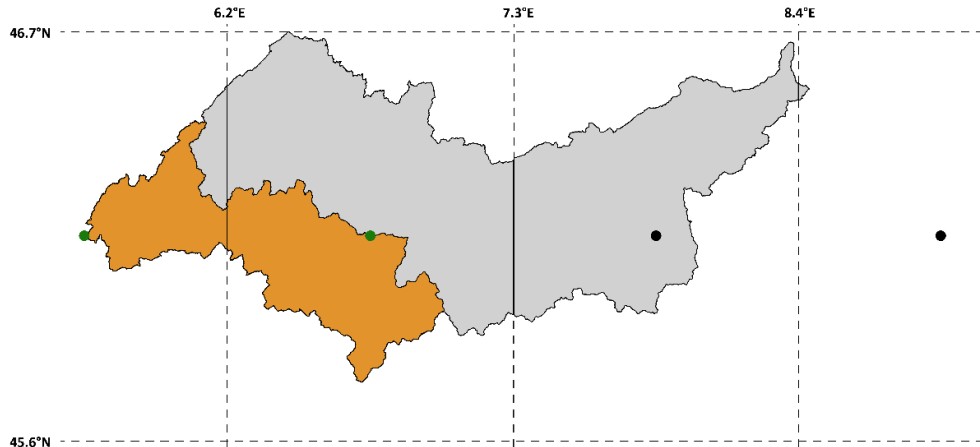

**Figure A4.** Location of the different ERA-20C grid points used to calculate the catchment precipitation for the period 1923-2010. The orange area corresponds to the A+V catchment, the green dots correspond to the two ERA-20C grid points used to calculate the catchment precipitation (the Thiessen polygons associated are shown). The grey area corresponds to the Geneva catchment, the two black and the two green dots correspond to the four ERA-20C grid points used to calculate the catchment precipitation (the Thiessen polygons are shown). Green dots represent ERA-20C grid points both used for the calculation of the A+V and Geneva catchment precipitation.

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
