# Peer review of "Is Precipitation the Main Trigger of Medium-Magnitude Floods in Large Alpine Catchments?"

_water, doi:10.3390/w11122507_

Round 1

Reviewer 1 Report

General comments

The idea behind this study is very interesting and promising, based on recent suggestions from the literature linking atmospheric synoptic indices to flood-relevant precipitation indices. It is certainly worth pursuing the further links of circulation patterns with specific flooding behavior. The river catchments are very well characterized, and the case study seems to be carried out carefully. The strong buffering influence of the Geneva lake is well accounted for and wisely the analysis is limited to the Arve and Valserine sub-catchments. Similar care is put in handling the reanalysis data and in presenting the results. I suggest that the paper should be considered for publication after a number of main points are adequately addressed, and after some aspects of the presentation and writing are improved. Also, English style and syntax is in need of more careful revision.

Some main suggestions:

The authors have placed a lot of emphasis, starting with the abstract, on the aspect of the link between flooding (discharge) and descriptors/metrics of synoptic atmospheric circulation states. Although this forms a solid rationale for carrying out the study, this is not in any way part of the work. I suggest contextualizing this better and paying attention to framing, mainly in the introduction, would prevent creating misplaced expectations for the paper in the reader.

By ‘flood’ here you mean extreme river discharge, without consideration of presence of water outside of the river banks (save from the 1990 event, for which you report damage). This is not the most common definition of floods, as used, e.g., in the natural disasters and flood management literature, nor in the common parlance. If you wish to be relevant to a wider community, please either clarify very early (also abstract) your definition, or re-word to ‘extreme discharge’.

Specific comments:

The title is worded somewhat strangely. I am not mother-tongue English, but why not simply ‘Are high-magnitude floods in xxx mainly triggered…?’ or ‘Is precipitation the main trigger…’

Abstract

A number of syntax mistakes need to be corrected.

The third sentence sounds very promising, but the reader is left to wonder about what ‘atmospheric processes’ we talk about.

The objectives need to be reformulated to be clearer.

L 17: there is no discharge from ERA 20C, as far as I know: what do you mean in this sentence?

L 18: I am not sure that a large portion of the flood-risk community would agree with high-magnitude floods being defined as 3-year events. I see you make this correspond to the 99.9 percentile of daily discharge (most daily discharge in 1000 days), but please document your choice of terminology, or change ‘high-magnitude’ for something else that’s more in line with the usage of relevant communities.

L 20: Directly or indirectly, precipitation can be seen as the only driver of river floods. I am sure you imply snow melt as the other main factor, but please clarify this set of concepts (do you also include ice-jam floods, for example?).

L 20: half of the ‘high-magnitude’ floods? More in general, I think it’s hard to see the main point that is made in this sentence. Consequently, the closing sentence remains also potentially unclear.

L 21: 20 years?

Introduction

First sentence is a bit empty.

L 37: what is vertical humidity?

L 42: I am not sure what you intend with extreme precipitation sequences. Also, do you rely on assumptions or on hypotheses? In general I would say hypotheses are there for the testing, assumption for the using.

L 48-49: rephrase, repetition of role. Also, this is not really clear. I would say that the predictive power of atmospheric/synoptic indicators/indices may be limited in such regions by confounding effect of specific hydrological processes like snowmelt. Or something along these lines.

L 50: to what extent (also elsewhere).

Why is not your objective to determine the relative explanatory power of each factor, rather than focusing a priori on precipitation only? By the way, it seems to me that the objective formulated in line 50 is not the same as the aim formulated in line 45. Please clarify.

L 51-54: this is confusing to me. I expected that you were going to rely on synoptic atmospheric metrics, but instead of investigating links between those and floods, it seems you will look at links between precipitation (which is poorly represented in GCMs) and floods.

L 55-57: ‘complex’ used twice in a row. Please use this word sparingly.

L 64: I don’t think your study should support anything yet, before presenting its results.

L 75: I am not sure you can ‘perform’ a typology.

What does section 5 do?

Section 2

L 83: I don’t think 1 100 a recognized way to write 1,100 or 1100.

L 91: units should be written properly.

L 174: why do you need to specify that you call ‘Geneva catchment precipitation’ ‘Geneva catchment precipitation’?

L 181 on: from strong correlation on overall precipitation does not result that also extreme precipitation is strongly correlated, please clarify the behavior specifically for extreme precipitation (e.g., which index?)

L 191 on: this explanation of the merit of the reanalysis dataset is not clear, please rephrase. Also, don’t you use it because it contains the mentioned metrics of synoptic atmospheric features, unlike the local rain gauges?

L 208 on: I am not entirely sure I follow correctly. It may help if gauges are always named the same across the paper. For example, don’t call the same thing either Rhone@HDI of Geneva catchement. I understand that the earlier is a proxy for the latter (right?), but it becomes confusing. Alternatively, always state to which gauge you refer. For example, in line 212 you compare a catchment to a gauge, which is confusing. Also, don’t go back to calling the A+V catchment with the full river names.

L 223: time of concentration? Maybe time of peak precipitation?

L 231: temporality is maybe here the lead time?

Fig. 2 caption. I would suggest to rephrase for clarity. Also, make sure the difference between panel a and b is immediately clear, e.g., by avoiding repeating anything that is common to both panels. The arrow may be not necessary, as the reader can clearly see where the highest percentiles are reached

L 242 on: related to fig 2 and caption, I think this explanation could be made clearer. At least, I have problems following confidently what you mean. E.g., when referring to catchment precipitation, you refer to the mean of the daily precipitation values?

L 266: median of what? I guess (daily) precipitation percentiles, but it’s good to be explicit and not assume the reader has your train of thought.

L 267: which both datasets, weather station and reanalysis I guess?

L 269: one duration or actually two durations?

L 272-275: this is no explanation for the choice of 2-day duration.

L 279: I don’t think I see how you determined the ‘temporality’ (lead time) that you associate with the 2-day and the 7-day duration descriptors.

L 298: I assume these are all average daily values of discharge, but please make explicit.

L 305: punctuation.

L 308 on: I would try to write this part more clearly. If I understand the method correctly, it clusters floods based on their mutual proximity in a 3-dimensional space defined by the three selected descriptors. But I may well be wrong. What is inertia here? Also, I don’t think you explained why you retained 3 and 4 types of flood. Greatest intra-class inertia would have been displayed also by the 2 and 3 groups, e.g. Linked to this: I don’t think you have reported on whether the 16 events from the weather station series are the same in the later part of the reanalysis series; that would be interesting.

Fig. 4 is not readable

L 345: I’d call this flood type 4, in keeping with the rest.

L 364: if you write ‘four’ in letters, maybe you avoid confusion with flood type 4.

Please consider, if you agree, condensing the parts that describe and point to the differences between the types of floods. As it is, I am afraid that many readers may skip it and only refer to Fig. 4 which seems to sum most traits up.

Discussion and conclusions

The line of discussion seems generally valid, and the comparison to similar approaches and results from publications are extremely careful. Figure 7 is very useful (as figure 6), and probably also table 2 is helpful. I refrain from making detailed comments on it, and just remark that: 1) in my opinion the discussion is very long, and may be condensed with some effort to the most salient points from the results and literature, in view of the paper’s objectives; 2) given the spirit of the study, of serving as preliminary test to then explore the use of circulation metrics as predictors of flooding, I expected a more explicit link with those metrics, maybe suggesting which ones could be linked with precipitation sequences that you have shown to be involved in a significant proportion of (high-magnitude) floods, or maybe indicating in slightly more detail which next steps should be taken towards setting up a methodology for Alpine flood forecast based on weather forecast or on GCM-based long term projected changes in atmospheric circulation.

Reviewer 2 Report

There are numerous minor English language problems that reduce the ease of comprehension and the enjoyment of reading.  I recommend spending some time examining the sentence grammar, in particular the verb tenses.  When in doubt, simplify the text.  Some editing support would greatly enhance the readability.  

Additionally, I believe there is a scope problem.  It seems that you set out to just look at the ability of rainfall to predict flooding but got distracted with a typology of floods.  I would recommend a serious reorganization / framing of the question from the start of the paper to set up the reader for the current content.  

I made more detailed comments in the manuscripts.  Unfortunately, it was less in my speciality as I had hoped so I cannot comment on the more technical aspects (methods and techniques).  I do think that if the new scope is embraced, correlations with some of the mentioned "hydrological" variables would be interesting, i.e. snow cover, antecedent moisture, etc. 

Round 2

Reviewer 2 Report

You adequately addressed those of my comments that you decided to address.  As you know, the language and writing still can be improved.